# Implicit Regularization in Matrix Factorization

**Suriya Gunasekar**
TTI at Chicago
suriya@ttic.edu

**Blake Woodworth**
TTI at Chicago
blake@ttic.edu

**Srinadh Bhojanapalli**
TTI at Chicago
srinadh@ttic.edu

**Behnam Neyshabur**
TTI at Chicago
behnam@ttic.edu

**Nathan Srebro**
TTI at Chicago
nati@ttic.edu

## Abstract

We study implicit regularization when optimizing an underdetermined quadratic objective over a matrix $X$ with gradient descent on a factorization of $X$. We conjecture and provide empirical and theoretical evidence that with small enough step sizes and initialization close enough to the origin, gradient descent on a full dimensional factorization converges to the minimum nuclear norm solution.

## 1 Introduction

When optimizing underdetermined problems with multiple global minima, the choice of optimization algorithm can play a crucial role in biasing us toward a specific global minima, even though this bias is not explicitly specified in the objective or problem formulation. For example, using gradient descent to optimize an unregularized, underdetermined least squares problem would yield the minimum Euclidean norm solution, while using coordinate descent or preconditioned gradient descent might yield a different solution. Such implicit bias, which can also be viewed as a form of regularization, can play an important role in learning.

In particular, implicit regularization has been shown to play a crucial role in training deep models [14, 13, 18, 11]: deep models often generalize well even when trained purely by minimizing the training error without any explicit regularization, and when there are more parameters than samples and the optimization problem is underdetermined. Consequently, there are many zero training error solutions, all global minima of the training objective, many of which generalize badly. Nevertheless, our choice of optimization algorithm, typically a variant of gradient descent, seems to prefer solutions that do generalize well. This generalization ability cannot be explained by the capacity of the explicitly specified model class (namely, the functions representable in the chosen architecture). Instead, it seems that the optimization algorithm biases us toward a "simple" model, minimizing some implicit "regularization measure", and that generalization is linked to this measure. But what are the regularization measures that are implicitly minimized by different optimization procedures?

As a first step toward understanding implicit regularization in complex models, in this paper we carefully analyze implicit regularization in matrix factorization models, which can be viewed as two-layer networks with linear transfer. We consider gradient descent on the entries of the factor matrices, which is analogous to gradient descent on the weights of a multilayer network. We show how such an optimization approach can indeed yield good generalization properties even when the problem is underdetermined. We identify the implicit regularizer as the *nuclear norm*, and show that even when we use a full dimensional factorization, imposing no constraints on the factored matrix, optimization by gradient descent on the factorization biases us toward the minimum nuclear norm solution. Our empirical study leads us to conjecture that with small step sizes and initialization close

to zero, gradient descent converges to the minimum nuclear norm solution, and we provide empirical and theoretical evidence for this conjecture, proving it in certain restricted settings.

## 2  Factorized Gradient Descent for Matrix Regression

We consider least squares objectives over matrices $X \in \mathbb{R}^{n \times n}$ of the form:

$$\min_{X \succeq 0} F(X) = \|\mathcal{A}(X) - y\|_2^2. \tag{1}$$

where $\mathcal{A} : \mathbb{R}^{n \times n} \to \mathbb{R}^m$ is a linear operator specified by $\mathcal{A}(X)_i = \langle A_i, X \rangle$, $A_i \in \mathbb{R}^{n \times n}$, and $y \in \mathbb{R}^m$. Without loss of generality, we consider only symmetric positive semidefinite (p.s.d.) $X$ and symmetric linearly independent $A_i$ (otherwise, consider optimization over a larger matrix $\begin{bmatrix} W & X \\ X^\top & Z \end{bmatrix}$ with $\mathcal{A}$ operating symmetrically on the off-diagonal blocks). In particular, this setting covers problems including matrix completion (where $A_i$ are indicators, [5]), matrix reconstruction from linear measurements [15] and multi-task training (where each column of $X$ is a predictor for a different task and $A_i$ have a single non-zero column, [2, 1]).

We are particularly interested in the regime where $m \ll n^2$, in which case (1) is underdetermined with many global minima satisfying $\mathcal{A}(X) = y$. For such underdetermined problems, merely minimizing (1) cannot ensure recovery (in matrix completion or recovery problems) or generalization (in prediction problems). For example, in a matrix completion problem (without diagonal observations), we can minimize (1) by setting all non-diagonal unobserved entries to zero, or to any arbitrary value.

Instead of working on $X$ directly, we will study a factorization $X = UU^\top$. We can write (1) equivalently as optimization over $U$ as,

$$\min_{U \in \mathbb{R}^{n \times d}} f(U) = \|\mathcal{A}(UU^\top) - y\|_2^2. \tag{2}$$

When $d < n$, this imposes a constraint on the rank of $X$, but we will be mostly interested in the case $d = n$, under which no additional constraint is imposed on $X$ (beyond being p.s.d.) and (2) is equivalent to (1). Thus, if $m \ll n^2$, then (2) with $d = n$ is similarly underdetermined and can be optimized in many ways – estimating a global optima cannot ensure generalization (e.g. imputing zeros in a matrix completion objective). Let us investigate what happens when we optimize (2) by gradient descent on $U$.

To simulate such a matrix reconstruction problem, we generated $m \ll n^2$ random measurement matrices and set $y = \mathcal{A}(X^*)$ according to some planted $X^* \succeq 0$. We minimized (2) by performing gradient descent on $U$ to convergence, and then measured the relative reconstruction error $\|X - X^*\|_F / \|X^*\|_F$ for $X = UU^\top$. Figure 1 shows the normalized training objective and reconstruction error as a function of the dimensionality $d$ of the factorization, for different initialization and step-size policies, and three different planted $X^*$.

First, we see that (for sufficiently large $d$) gradient descent indeed finds a global optimum, as evidenced by the training error (the optimization objective) being zero. This is not surprising since with large enough $d$ this non-convex problem has no spurious local minima [4, 9] and gradient descent converges almost surely to a global optima [12]; there has also been recent work establishing conditions for global convergence for low $d$ [3, 7].

The more surprising observation is that in panels $(a)$ and $(b)$, even when $d > m/n$, indeed even for $d = n$, we still get good reconstructions from the solution of gradient descent with initialization $U_0$ close to zero and small step size. In this regime, (2) is underdetermined and minimizing it does not ensure generalization. To emphasize this, we plot the reference behavior of a rank unconstrained global minimizer $X_{gd}$ obtained via projected gradient descent for (1) on the $X$ space. For $d < n$ we also plot an example of an alternate "bad" rank $d$ global optima obtained with an initialization based on SVD of $X_{gd}$ ('SVD Initialization').

When $d < m/n$, we understand how the low-rank structure can guarantee generalization [16] and reconstruction [10, 3, 7]. What ensures generalization when $d \gg m/n$? Is there a strong implicit regularization at play for the case of gradient descent on factor space and initialization close to zero?

Observing the nuclear norm of the resulting solutions plotted in Figure 2 suggests that gradient descent implicitly induces a low nuclear norm solution. This is the case even for $d = n$ when the factorization

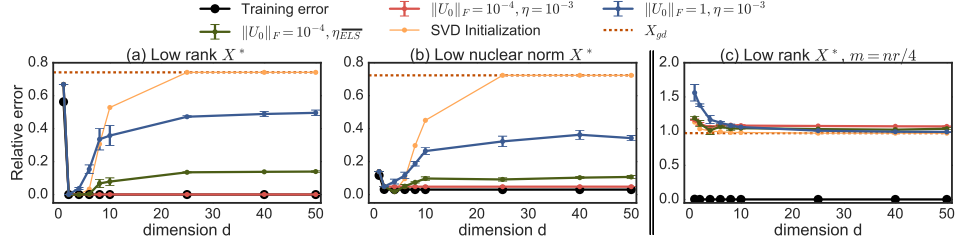

Figure 1: Reconstruction error of the global optima for $50 \times 50$ matrix reconstruction. *(Left)* $X^*$ is of rank $r = 2$ and $m = 3nr$; *(Center)* $X^*$ has a spectrum decaying as $O(1/k^{1.5})$ normalized to have $\|X^*\|_* = \sqrt{r}\|X^*\|_F$ for $r = 2$ and $m = 3nr$, and *(Right)* is a non-reconstructable setting where the number of measurements $m = nr/4$ is much smaller than the requirement to reconstruct a rank $r = 2$ matrix. The plots compare the reconstruction error of gradient descent on $U$ for different choices initialization $U_0$ and step size $\eta$, including fixed step-size and exact line search clipped for stability ($\eta_{\overline{ELS}}$). Additonally, the orange dashed reference line represents the performance of $X_{gd}$ – a rank unconstrained global optima obtained by projected gradient descent for (1) on $X$ space, and 'SVD-Initialization' is an example of an alternate rank $d$ global optima, where initialization $U_0$ is picked based on SVD of $X_{gd}$ and gradient descent is run on factor space with small stepsize. Training error behaves similarly in all these settings (zero for $d \geq 2$) and is plotted for reference. Results are averaged across 3 random initialization and (near zero) errorbars indicate the standard deviation.

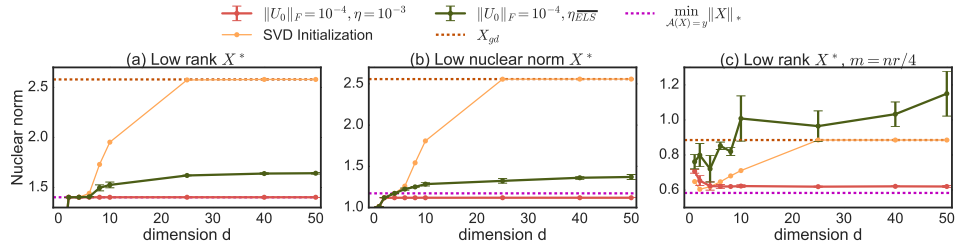

Figure 2: Nuclear norm of the solutions from Figure 1. In addition to the reference of $X_{gd}$ from Figure 1, the magenta dashed line (almost overlapped by the plot of $\|U\|_F = 10^{-4}, \eta = 10^{-3}$) is added as a reference for the (rank unconstrained) minimum nuclear norm global optima. The error bars indicate the standard deviation across 3 random initializations. We have dropped the plot for $\|U\|_F = 1, \eta = 10^{-3}$ to reduce clutter.

imposes no explicit constraints. Furthermore, we do not include any explicit regularization and optimization is run to convergence without any early stopping. In fact, we can see a clear bias toward low nuclear norm even in problems where reconstruction is not possible: in panel (c) of Figure 2 the number of samples $m = nr/4$ is much smaller than those required to reconstruct a rank $r$ ground truth matrix $X^*$. The optimization in (2) is highly underdetermined and there are many possible zero-error global minima, but gradient descent still prefers a lower nuclear norm solution. The emerging story is that gradient descent biases us to a low nuclear norm solution, and we already know how having low nuclear norm can ensure generalization [17, 6] and minimizing the nuclear norm ensures reconstruction [15, 5].

Can we more explicitly characterize this bias? We see that we do not always converge precisely to the minimum nuclear norm solution. In particular, the choice of step size and initialization affects which solution gradient descent converges to. Nevertheless, as we formalize in Section 3, we argue that when $U$ is full dimensional, the step size becomes small enough, and the initialization approaches zero, gradient descent will converge precisely to a minimum nuclear norm solution, i.e. to $\operatorname{argmin}_{X \succeq 0} \|X\|_*$ s.t. $\mathcal{A}(X) = y$.

## 3 Gradient Flow and Main Conjecture

The behavior of gradient descent with infinitesimally small step size is captured by the differential equation $\dot{U}_t := \frac{dU_t}{dt} = -\nabla f(U_t)$ with an initial condition for $U_0$. For the optimization in (2) this is

$$\dot{U}_t = -\mathcal{A}^*(\mathcal{A}(U_t U_t^\top) - y)U_t, \tag{3}$$

where $\mathcal{A}^* : \mathbb{R}^m \rightarrow \mathbb{R}^{n \times n}$ is the adjoint of $\mathcal{A}$ and is given by $\mathcal{A}^*(r) = \sum_i r_i A_i$. Gradient descent can be seen as a discretization of (3), and approaches (3) as the step size goes to zero.

The dynamics (3) define the behavior of the solution $X_t = U_t U_t^\top$ and using the chain rule we can verify that $\dot{X}_t = \dot{U}_t U_t^\top + U_t \dot{U}_t^\top = -\mathcal{A}^*(r_t) X_t - X_t \mathcal{A}^*(r_t)$, where $r_t = \mathcal{A}(X_t) - y$ is a vector of the residual. That is, even though the dynamics are defined in terms of specific factorization $X_t = U_t U_t^\top$, they are actually independent of the factorization and can be equivalently characterized as

$$\dot{X}_t = -\mathcal{A}^*(r_t) X_t - X_t \mathcal{A}^*(r_t). \tag{4}$$

We can now define the limit point $X_\infty(X_{\text{init}}) := \lim_{t \rightarrow \infty} X_t$ for the factorized gradient flow (4) initialized at $X_0 = X_{\text{init}}$. We emphasize that these dynamics are very different from the standard gradient flow dynamics of (1) on $X$, corresponding to gradient descent on $X$, which take the form $\dot{X}_t = -\nabla F(X_t) = -\mathcal{A}^*(r_t)$.

Based on the preliminary experiments in Section 2 and a more comprehensive numerical study discussed in Section 5, we state our main conjecture as follows:

**Conjecture.** *For any full rank $X_{init}$, if $\widehat{X} = \lim_{\alpha \rightarrow 0} X_\infty(\alpha X_{init})$ exists and is a global optima for (1) with $\mathcal{A}(\widehat{X}) = y$, then $\widehat{X} \in \operatorname{argmin}_{X \succeq 0} \|X\|_*$ s.t. $\mathcal{A}(X) = y$.*

Requiring a full-rank initial point demands a full dimensional $d = n$ factorization in (2). The assumption of global optimality in the conjecture is generally satisfied: for almost all initializations, gradient flow will converge to a local minimizer [12], and when $d = n$ any such local minimizer is also global minimum [9]. Since we are primarily concerned with underdetermined problems, we expect the global optimum to achieve zero error, i.e. satisfy $\mathcal{A}(X) = y$. We already know from these existing literature that gradient descent (or gradient flow) will generally converge to *a* solution satisfying $\mathcal{A}(X) = y$; the question we address here is *which* of those solutions will it converge to.

The conjecture implies the same behavior for asymmetric factorization as $X = UV^\top$ with gradient flow on $(U, V)$, since this is equivalent to gradient flow on the p.s.d. factorization of $\begin{bmatrix} W & X \\ X^\top & Z \end{bmatrix}$.

## 4 Theoretical Analysis

We will prove our conjecture for the special case where the matrices $A_i$ commute, and discuss the more challenging non-commutative case. But first, let us begin by reviewing the behavior of straight-forward gradient descent on $X$ for the convex problem in (1).

**Warm up:** Consider gradient descent updates on the original problem (1) in $X$ space, ignoring the p.s.d. constraint. The gradient direction $\nabla F(X) = \mathcal{A}^*(\mathcal{A}(X) - y)$ is always spanned by the $m$ matrices $A_i$. Initializing at $X_{\text{init}} = 0$, we will therefore always remain in the $m$-dimensional subspace $\mathcal{L} = \{X = \mathcal{A}^*(s) | s \in \mathbb{R}^m\}$. Now consider the optimization problem $\min_X \|X\|_F^2$ s.t. $\mathcal{A}(X) = y$. The KKT optimality conditions for this problem are $\mathcal{A}(X) = y$ and $\exists \nu$ s.t. $X = \mathcal{A}^*(\nu)$. As long as we are in $\mathcal{L}$, the second condition is satisfied, and if we converge to a zero-error global minimum, then the first condition is also satisfied. Since gradient descent stays on this manifold, this establishes that if gradient descent converges to a zero-error solution, it is the minimum Frobenius norm solution.

**Getting started: $m = 1$** Consider the simplest case of the factorized problem when $m = 1$ with $A_1 = A$ and $y_1 = y$. The dynamics of (4) are given by $\dot{X}_t = -r_t(AX_t + X_t A)$, where $r_t$ is simply a scalar, and the solution for $X_t$ is given by, $X_t = \exp(s_t A) X_0 \exp(s_t A)$ where $s_T = -\int_0^T r_t dt$. Assuming $\widehat{X} = \lim_{\alpha \rightarrow 0} X_\infty(\alpha X_0)$ exists and $\mathcal{A}(\widehat{X}) = y$, we want to show $\widehat{X}$ is an optimum for the following problem

$$\min_{X \succeq 0} \|X\|_* \quad \text{s.t.} \ \mathcal{A}(X) = y. \tag{5}$$

The KKT optimality conditions for (5) are:

$$\exists \nu \in \mathbb{R}^m \text{ s.t.} \qquad \mathcal{A}(X) = y \qquad X \succeq 0 \qquad \mathcal{A}^*(\nu) \preceq I \qquad (I - \mathcal{A}^*(\nu))X = 0 \tag{6}$$

We already know that the first condition holds, and the p.s.d. condition is guaranteed by the factorization of $X$. The remaining complementary slackness and dual feasibility conditions effectively require

that $\widehat{X}$ is spanned by the top eigenvector(s) of $A$. Informally, looking to the gradient flow path above, for any non-zero $y$, as $\alpha \to 0$ it is necessary that $|s_\infty| \to \infty$ in order to converge to a global optima, thus eigenvectors corresponding to the top eigenvalues of $A$ will dominate the span of $X_\infty(\alpha X_{\text{init}})$.

**What we can prove: Commutative $\{\mathbf{A_i}\}_{\mathbf{i \in [m]}}$** The characterization of the the gradient flow path from the previous section can be extended to arbitrary $m$ in the case that the matrices $A_i$ commute, i.e. $A_i A_j = A_j A_i$ for all $i, j$. Defining $s_T = -\int_0^T r_t dt$ – a vector integral, we can verify by differentiating that solution of (4) is

$$X_t = \exp\left(\mathcal{A}^*(s_t)\right) X_0 \exp\left(\mathcal{A}^*(s_t)\right) \tag{7}$$

**Theorem 1.** *In the case where matrices $\{A_i\}_{i=1}^m$ commute, if $\widehat{X} = \lim_{\alpha \to 0} X_\infty(\alpha I)$ exists and is a global optimum for* (1) *with $\mathcal{A}(\widehat{X}) = y$, then $\widehat{X} \in \operatorname{argmin}_{X \succeq 0} \|X\|_*$ s.t. $\mathcal{A}(X) = y$.*

*Proof.* It suffices to show that such a $\widehat{X}$ satisfies the complementary slackness and dual feasibility KKT conditions in (6). Since the matrices $A_i$ commute and are symmetric, they are simultaneously diagonalizable by a basis $v_1, .., v_n$, and so is $\mathcal{A}^*(s)$ for any $s \in \mathbb{R}^m$. This implies that for any $\alpha$, $X_\infty(\alpha I)$ given by (7) and its limit $\widehat{X}$ also have the same eigenbasis. Furthermore, since $X_\infty(\alpha I)$ converges to $\widehat{X}$, the scalars $v_k^\top X_\infty(\alpha I) v_k \to v_k^\top \widehat{X} v_k$ for each $k \in [n]$. Therefore, $\lambda_k(X_\infty(\alpha I)) \to \lambda_k(\widehat{X})$, where $\lambda_k(\cdot)$ is defined as the eigenvalue corresponding to eigenvector $v_k$ and *not* necessarily the $k^{\text{th}}$ largest eigenvalue.

Let $\beta = -\log \alpha$, then using $X_0 = e^{-\beta} I$ in (7), $\lambda_k(X_\infty(\alpha I)) = \exp(2\lambda_k(\mathcal{A}^*(s_\infty(\beta))) - 2\beta)$. For all $k$ such that $\lambda_k(\widehat{X}) > 0$, by the continuity of log, we have

$$2\lambda_k(\mathcal{A}^*(s_\infty(\beta))) - 2\beta - \log \lambda_k(\widehat{X}) \to 0 \implies \lambda_k\left(\mathcal{A}^*\left(\frac{s_\infty(\beta)}{\beta}\right)\right) - 1 - \frac{\log \lambda_k(\widehat{X})}{2\beta} \to 0. \tag{8}$$

Defining $\nu(\beta) = s_\infty(\beta)/\beta$, we conclude that for all $k$ such that $\lambda_k(\widehat{X}) \neq 0$, $\lim_{\beta \to \infty} \lambda_k(\mathcal{A}^*(\nu(\beta))) = 1$. Similarly, for each $k$ such that $\lambda_k(\widehat{X}) = 0$,

$$\exp(2\lambda_k(\mathcal{A}^*(s_\infty(\beta))) - 2\beta) \to 0 \implies \exp(\lambda_k(\mathcal{A}^*(\nu(\beta))) - 1)^{2\beta} \to 0. \tag{9}$$

Thus, for every $\epsilon \in (0, 1]$, for sufficiently large $\beta$

$$\exp(\lambda_k(\mathcal{A}^*(\nu(\beta))) - 1) < \epsilon^{\frac{1}{2\beta}} < 1 \implies \lambda_k(\mathcal{A}^*(\nu(\beta))) < 1. \tag{10}$$

Therefore, we have shown that $\lim_{\beta \to \infty} \mathcal{A}^*(\nu(\beta)) \preceq I$ and $\lim_{\beta \to \infty} \mathcal{A}^*(\nu(\beta))\widehat{X} = \widehat{X}$ establishing the optimality of $\widehat{X}$ for (5). $\qquad\square$

Interestingly, and similarly to gradient descent on $X$, this proof does not exploit the particular form of the "control" $r_t$ and only relies on the fact that the gradient flow path stays within the manifold

$$\mathcal{M} = \{X = \exp\left(\mathcal{A}^*(s)\right) X_{\text{init}} \exp\left(\mathcal{A}^*(s)\right) \mid s \in \mathbb{R}^m\}. \tag{11}$$

Since the $A_i$'s commute, we can verify that the tangent space of $\mathcal{M}$ at a point $X$ is given by $T_X \mathcal{M} = \operatorname{Span}\{A_i X + X A_i\}_{i \in [m]}$, thus gradient flow will always remain in $\mathcal{M}$. For any control $r_t$ such that following $\dot{X}_t = -\mathcal{A}^*(r_t)X_t - X_t \mathcal{A}^*(r_t)$ leads to a zero error global optimum, that optimum will be a minimum nuclear norm solution. This implies in particular that the conjecture extends to gradient flow on (2) even when the Euclidean norm is replaced by certain other norms, or when only a subset of measurements are used for each step (such as in stochastic gradient descent).

However, unlike gradient descent on $X$, the manifold $\mathcal{M}$ is not flat, and the tangent space at each point is different. Taking finite length steps, as in gradient descent, would cause us to "fall off" of the manifold. To avoid this, we must take infinitesimal steps, as in the gradient flow dynamics.

In the case that $X_{\text{init}}$ and the measurements $A_i$ are diagonal matrices, gradient descent on (2) is equivalent to a vector least squares problem, parametrized in terms of the square root of entries:

**Corollary 2.** *Let $x_\infty(x_{init})$ be the limit point of gradient flow on $\min_{u \in \mathbb{R}^n} \|Ax(u) - y\|_2^2$ with initialization $x_{init}$, where $x(u)_i = u_i^2$, $A \in \mathbb{R}^{m \times n}$ and $y \in \mathbb{R}^m$. If $\widehat{x} = \lim_{\alpha \to 0} x_\infty(\alpha \vec{1})$ exists and $A\widehat{x} = y$, then $\widehat{x} \in \operatorname{argmin}_{x \in \mathbb{R}_+^m} \|x\|_1$ s.t. $Ax = y$.*

**The plot thickens: Non-commutative $\{\mathbf{A_i}\}_{\mathbf{i} \in [\mathbf{m}]}$**    Unfortunately, in the case that the matrices $A_i$ do not commute, analysis is much more difficult. For a matrix-valued function $F$, $\frac{\mathrm{d}}{\mathrm{d}t} \exp(F_t)$ is equal to $\dot{F}_t \exp(F_t)$ only when $\dot{F}_t$ and $F_t$ commute. Therefore, (7) is no longer a valid solution for (4). Discretizing the solution path, we can express the solution as the "time ordered exponential":

$$X_t = \lim_{\epsilon \to 0} \left( \prod_{\tau = t/\epsilon}^{1} \exp\left(-\epsilon \mathcal{A}^*(r_{\tau\epsilon})\right) \right) X_0 \left( \prod_{\tau=1}^{t/\epsilon} \exp\left(-\epsilon \mathcal{A}^*(r_{\tau\epsilon})\right) \right), \qquad (12)$$

where the order in the products is important. If $A_i$ commute, the product of exponentials is equal to an exponential of sums, which in the limit evaluates to the solution in (7). However, since in general $\exp(A_1) \exp(A_2) \neq \exp(A_1 + A_2)$, the path (12) is *not* contained in the manifold $\mathcal{M}$ defined in (11).

It is tempting to try to construct a new manifold $\mathcal{M}'$ such that $\mathrm{Span}\left\{A_i X + X A_i\right\}_{i \in [m]} \subseteq T_X \mathcal{M}'$ and $X_0 \in \mathcal{M}'$, ensuring the gradient flow remains in $\mathcal{M}'$. However, since $A_i$'s do not commute, by combining infinitesimal steps along different directions, it is possible to move (very slowly) in directions that are *not* of the form $\mathcal{A}^*(s) X + X \mathcal{A}^*(s)$ for any $s \in \mathbb{R}^m$. The possible directions of movements indeed corresponds to the Lie algebra defined by the closure of $\{A_i\}_{i=1}^m$ under the commutator operator $[A_i, A_j] := A_i A_j - A_j A_i$. Even when $m = 2$, this closure will generally encompass *all* of $\mathbb{R}^{n \times n}$, allowing us to approach any p.s.d. matrix $X$ with some (wild) control $r_t$. Thus, we cannot hope to ensure the KKT conditions for an arbitrary control as we did in the commutative case — it is necessary to exploit the structure of the residuals $\mathcal{A}(X_t) - y$ in some way.

Nevertheless, in order to make finite progress moving along a commutator direction like $[A_i, A_j] X_t + X_t [A_i, A_j]^\top$, it is necessary to use an extremely non-smooth control, e.g., looping $1/\epsilon^2$ times between $\epsilon$ steps in the directions $A_i, A_j, -A_i, -A_j$, each such loop making an $\epsilon^2$ step in the desired direction. We expect the actual residuals $r_t$ to behave much more smoothly and that for smooth control the non-commutative terms in the expansion of the time ordered exponential (12) are asymptotically lower order then the direct term $\mathcal{A}^*(s)$ (as $X_{\mathrm{init}} \to 0$). This is indeed confirmed numerically, both for the actual residual controls of the gradient flow path, and for other random controls.

## 5   Empirical Evidence

Beyond the matrix reconstruction experiments of Section 2, we also conducted experiments with similarly simulated matrix completion problems, including problems where entries are sampled from power-law distributions (thus not satisfying incoherence), as well as matrix completion problem on non-simulated Movielens data. In addition to gradient descent, we also looked more directly at the gradient flow ODE (3) and used a numerical ODE solver provided as part of `SciPy` [8] to solve (3). But we still uses a finite (non-zero) initialization. We also emulated staying on a valid "steering path" by numerically approximating the time ordered exponential of 12 — for a finite discretization $\eta$, instead of moving linearly in the direction of the gradient $\nabla f(U)$ (like in gradient descent), we multiply $X_t$ on right and left by $e^{-\eta \mathcal{A}^*(r_t)}$. The results of these experiments are summarized in Figure 3.

In these experiments, we again observe trends similar to those in Section 2. In some panels in Figure 3, we do see a discernible gap between the minimum nuclear norm global optima and the nuclear norm of the gradient flow solution with $\|U_0\|_F = 10^{-4}$. This discrepancy could either be due to starting at a non-limit point of $U_0$, or numerical issue arising from approximations to the ODE, or it could potentially suggest a weakening of the conjecture. Even if the later case were true, the experiments so far provide strong evidence for atleast approximate versions of our conjecture being true under a wide range of problems.

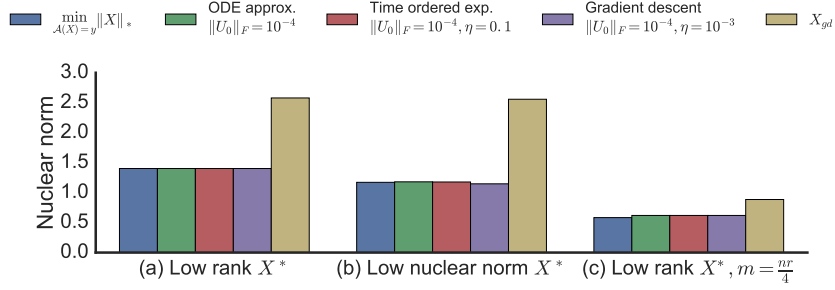

(i) Gaussian random measurements. We report the nuclear norm of the gradient flow solutions from three different approximations to (3) – numerical ODE solver (*ODE approx.*), time ordered exponential specified in (12) (*Time ordered exp.*) and standard gradient descent with small step size (*Gradient descent*). The nuclear norm of the solution from gradient descent on $X$ space – $X_{gd}$ and the minimum nuclear norm global minima are provided as references. In $(a)$ $X^*$ is rank $r$ and $m = 3nr$, in $(b)$ $X^*$ has a decaying spectrum with $\|X^*\|_* = \sqrt{r}\|X^*\|_F$ and $m = 3nr$, and in $(c)$ $X^*$ is rank $r$ with $m = nr/4$, where $n = 50, r = 2$.

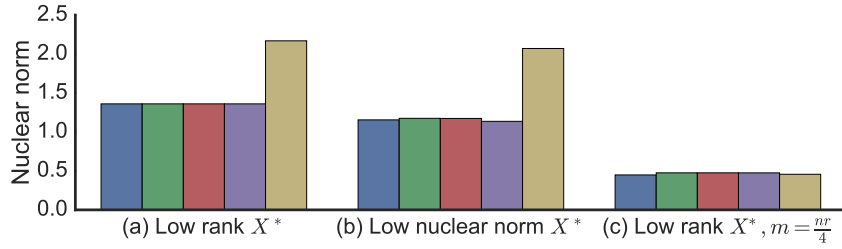

(ii) Uniform matrix completion: $\forall i$, $A_i$ measures a uniform random entry of $X^*$. Details on $X^*$, number of measurements, and the legends follow Figure3-(i).

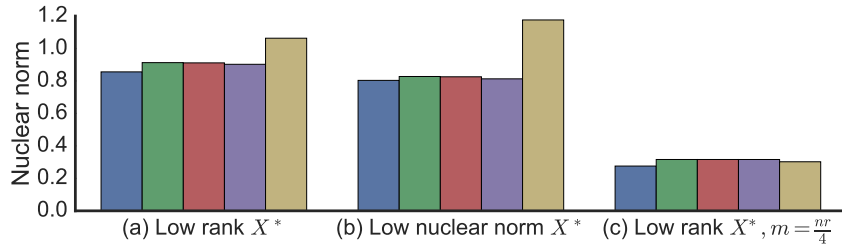

(iii) Power law matrix completion: $\forall i$, $A_i$ measures a random entry of $X^*$ chosen according to a power law distribution. Details on $X^*$, number of measurements, and the legends follow Figure3-(i).

|  | $\mathrm{argmin}_{\mathcal{A}(X)=y} \|X\|_*$ | Gradient descent $\|U_0\|_F = 10^{-3}, \eta = 10^{-2}$ | $X_{gd}$ |
|---|---|---|---|
| Test Error | 0.2880 | 0.2631 | 1.000 |
| Nuclear norm | 8391 | 8876 | 20912 |

(iv) Benchmark movie recommendation dataset — Movielens 100k. The dataset contains $\sim$ 100k ratings from $n_1 = 943$ users on $n_2 = 1682$ movies. In this problem, gradient updates are performed on the asymmetric matrix factorization space $X = UV^\top$ with dimension $d = \min(n_1, n_2)$. The training data is completely fit to have $< 10^{-2}$ error. Test error is computed on a held out data of 10 ratings per user. Here we are not interested in the recommendation performance (test error) itself but on observing the bias of gradient flow with initialization close to zero to return a low nuclear norm solution — the test error is provided merely to demonstrate the effectiveness of such a bias in this application. Also, due to the scale of the problem, we only report a coarse approximation of the gradient flow 3 from gradient descent with $\|U_0\|_F = 10^{-3}, \eta = 10^{-2}$.

Figure 3: Additional matrix reconstruction experiments

**Exhaustive search** Finally, we also did experiments on an exhaustive grid search over small problems, capturing essentially all possible problems of this size. We performed an exhaustive grid search for matrix completion problem instances in symmetric p.s.d. $3 \times 3$ matrices. With $m = 4$, there are 15 unique masks or $\{A_i\}_{i \in [4]}$'s that are valid symmetric matrix completion observations.

For each mask, we fill the $m = 4$ observations with all possible combinations of 10 uniformly spaced values in the interval $[-1, 1]$. This gives us a total of $15 \times 10^4$ problem instances. Of these problems instances, we discard the ones that do not have a valid PSD completion and run the ODE solver on every remaining instance with a random $U_0$ such that $\|U_0\|_F = \bar{\alpha}$, for different values of $\bar{\alpha}$. Results on the deviation from the minimum nuclear norm are reported in Figure 4. For small $\bar{\alpha} = 10^{-5}, 10^{-3}$, most of instances of our grid search algorithm returned solutions with near minimal nuclear norms, and the deviations are within the possibility of numerical error. This behavior also decays for $\bar{\alpha} = 1$.

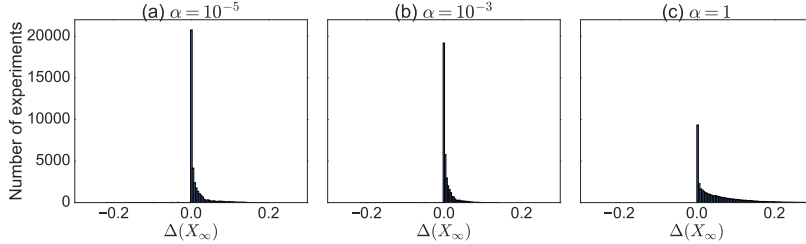

Figure 4: Histogram of relative sub-optimality of nuclear norm of $X_\infty$ in grid search experiments. We plot the histogram of $\Delta(X_\infty) = \frac{\|X_\infty\|_* - \|X_{\min}\|_*}{\|X_{\min}\|_*}$, where $\|X_{\min}\|_* = \min_{\mathcal{A}(X)=y} \|X\|_*$. The panels correspond to different values of norm of initialization $\bar{\alpha} = \|U_0\|_F$. **(Left)** $\bar{\alpha} = 10^{-5}$, **(Center)** $\bar{\alpha} = 10^{-3}$, and **(Right)** $\bar{\alpha} = 1$.

# 6    Discussion

It is becoming increasingly apparent that biases introduced by optimization procedures, especially for under-determined problems, are playing a key role in learning. Yet, so far we have very little understanding of the implicit biases associated with different non-convex optimization methods. In this paper we carefully study such an implicit bias in a two-layer non-convex problem, identify it, and show how even though there is no difference in the model class (problems (1) and (2) are equivalent when $d = n$, both with very high capacity), the non-convex modeling induces a potentially much more useful implicit bias.

We also discuss how the bias in the non-convex case is much more delicate then in convex gradient descent: since we are not restricted to a flat manifold, the bias introduced by optimization depends on the step sizes taken. Furthermore, for linear least square problems (i.e. methods based on the gradients w.r.t. $X$ in our formulation), any global optimization method that uses linear combination of gradients, including conjugate gradient descent, Nesterov acceleration and momentum methods, remains on the manifold spanned by the gradients, and so leads to the same minimum norm solution. This is not true if the manifold is curved, as using momentum or passed gradients will lead us to "shoot off" the manifold.

Much of the recent work on non-convex optimization, and matrix factorization in particular, has focused on global convergence: whether, and how quickly, we converge to a global minima. In contrast, we address the complimentary question of *which* global minima we converge to. There has also been much work on methods ensuring good matrix reconstruction or generalization based on structural and statistical properties. We do not assume any such properties, nor that reconstruction is possible or even that there is anything to reconstruct—for any problem of the form (1) we conjecture that (4) leads to the minimum nuclear norm solution. Whether such a minimum nuclear norm solution is good for reconstruction or learning is a separate issue already well addressed by the above literature.

We based our conjecture on extensive numerical simulations, with random, skewed, reconstructible, non-reconstructible, incoherent, non-incoherent, and and exhaustively enumerated problems, some of which is reported in Section 5. We believe our conjecture holds, perhaps with some additional technical conditions or corrections. We explain how the conjecture is related to control on manifolds and the time ordered exponential and discuss a possible approach for proving it.

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
