[Reviews · NeurIPS 2017]

Reviewer 1



This paper considers a foundational problem in machine learning: why do simple algorithms like gradient descent generalize well? Can we view these simple algorithms as implementing some kind of implicit regularization? The authors address this problem in the case of the simplest kind of non-convex learning model: gradient descent for matrix factorization. They (mostly) show that this method converges to the least nuclear norm solution. Importantly, they are honest about the limitations of their current theory, which currently requires commutativity of the measurement matrices. The questions asked by this paper are stronger than the answers. The authors seem well aware of the weaknesses of their results so far: * theory holds only for commutative case, which is not practically relevant * theory holds only in the continuous (gradient flow) limit * connection to deep learning is tenuous However, the questions (and partial answers) seem interesting enough to merit publication. It seems likely to this reviewer that these weaknesses are limitations of the technical tools currently available, rather than of the broader ideas presented in this paper. The numerical experiments, in particular, are thorough and support the conjecture. line 45: typo "deferent" figure 2 caption: missing period Conjecture: your experiments observe that the frobenius norm of the initial point seems to matter. State why this norm does not appear in your conjecture. line 160: where did the first equation come from? Also, I believe you have not defined s_\infty explicitly.

Reviewer 2



This paper studies an interesting observation: with small enough step sizes and initialization close enough to the origin, even for an underdetermined / unregularized problem of the form (1), gradient descent seems to find the solution with minimum nuclear norm. In other words, starting from an initialization with low nuclear norm, gradient descent seems to increase the norm of the iterates "just as needed". The authors state that conjecture and then proceed to prove it for the restricted case when the operator A is based on commutative matrices. The paper is pretty interesting and is a first step towards understanding the implicit regularization imposed by the dynamics of an optimization algorithm such as gradient descent. There seems to be many small errors in Figure 1 and 2 (see below). Detailed comments ----------------- Line 46: I think there are way earlier references than [16] for multi-task learning. I think it is a good idea to give credit to earlier references. Another example of model that would fall within this framework is factorization machines and their convex formulations. Line 63: I guess X = U U^T? X is not defined... Figure 1: - the figure does not indicate (a) and (b) (I assume they are the left and right plots) - what does the "training error" blue line refer to? - X_gd is mentioned in the caption but does not appear anywhere in the plot Figure 2: - I assume this is the nuclear norm of U U^T - X_gd (dotted black line) does not appear anywhere - It is a bit hard to judge how close the nuclear norms are to the magenta line without a comparison point - How big is the nuclear norm when d < 40? (not dislpayed) Line 218: I assume the ODE solver is used to solve Eq. (3)? Please clarify.

Reviewer 3



In this manuscript the authors present an interesting problem for the optimization community and in particular for machine learning applications, due to its implication. According to the authors, the proposed method obtain the minimization of the nuclear norm subject to linear constraints and considering only symmetric positive semidefinite matrices. First, a general conjecture is presented and then a particular case is proved. This is a theoretical work and the main contribution is to prove the previous particular case, and also provide an open problem to the community (the conjecture) which is very interesting. From my point of view, this is a working process manuscript, and some details should be completed. In the proof of theorem 1, which is the main contribution, there is a subtlety in the proof that should be considered. Which is the limit of s_t? or Which is the limit of s_t/\beta? Or, \eta(\beta) has limit? This should be answered because this was not considered in the proof, although it was mentioned in the previous section of the proof that |s_t | should tend to infinity when alpha goes to 0. But this should be proved.. The authors state that the proposal is more general. Please, give some insights or proof in these cases for example, what happens if the matrix is non symmetric or is not positive semidefinite…? The empirical evidence should be improved. The example with 3 by 3 matrices is too basic. The author should consider more practical examples or experiments.